# Minimizing a Submodular Function from Samples

**Eric Balkanski**
Harvard University
ericbalkanski@g.harvard.edu

**Yaron Singer**
Harvard University
yaron@seas.harvard.edu

## Abstract

In this paper we consider the problem of minimizing a submodular function from
training data. Submodular functions can be efficiently minimized and are conse-
quently heavily applied in machine learning. There are many cases, however, in
which we do not know the function we aim to optimize, but rather have access
to training data that is used to learn it. In this paper we consider the question of
whether submodular functions can be minimized when given access to its training
data. We show that even learnable submodular functions cannot be minimized
within any non-trivial approximation when given access to polynomially-many sam-
ples. Specifically, we show that there is a class of submodular functions with range
in $[0, 1]$ such that, despite being `PAC`-learnable and minimizable in polynomial-time,
no algorithm can obtain an approximation strictly better than $1/2 - o(1)$ using
polynomially-many samples drawn from any distribution. Furthermore, we show
that this bound is tight via a trivial algorithm that obtains an approximation of $1/2$.

## 1 Introduction

For well over a decade now, submodular minimization has been heavily studied in machine learning
(e.g. [SK10, JB11, JLB11, NB12, EN15, DTK16]). This focus can be largely attributed to the fact
that if a set function $f : 2^N \to \mathbb{R}$ is submodular, meaning it has the following property of diminishing
returns: $f(S \cup \{a\}) - f(S) \geq f(T \cup \{a\}) - f(T)$ for all $S \subseteq T \subseteq N$ and $a \notin T$, then it can
be optimized efficiently: its minimizer can be found in time that is polynomial in the size of the
ground set $N$ [GLS81, IFF01]. In many cases, however, we do not know the submodular function,
and instead learn it from data (e.g. [BH11, IJB13, FKV13, FK14, Bal15, BVW16]). The question
we address in this paper is whether submodular functions can be (approximately) minimized when
the function is not known but can be learned from training data.

An intuitive approach for optimization from training data is to learn a surrogate function from training
data that predicts the behavior of the submodular function well, and then find the minimizer of the
surrogate learned and use that as a proxy for the true minimizer we seek. The problem however, is that
this approach does not generally guarantee that the resulting solution is close to the true minimum of
the function. One pitfall is that the surrogate may be non-submodular, and despite approximating the
true submodular function arbitrarily well, the surrogate can be intractable to minimize. Alternatively,
it may be that the surrogate is submodular, but its minimum is arbitrarily far from the minimum of
the true function we aim to optimize (see examples in Appendix A).

Since optimizing a surrogate function learned from data may generally result in poor approximations,
one may seek learning algorithms that are guaranteed to produce surrogates whose optima well-
approximate the true optima and are tractable to compute. More generally, however, it is possible that
there is some other approach for optimizing the function from the training samples, without learning
a model. Therefore, at a high level, the question is whether a reasonable number of training samples
suffices to minimize a submodular function. We can formalize this as *optimization from samples*.

**Optimization from samples.** We will say that a class of functions $\mathcal{F} = \{f : 2^N \to [0,1]\}$ is *$\alpha$-optimizable from samples over distribution $\mathcal{D}$* if for every $f \in \mathcal{F}$ and $\delta \in (0,1)$, when given $\text{poly}(|N|)$ i.i.d. samples $\{(S_i, f(S_i))\}_{i=1}^m$ where $S_i \sim \mathcal{D}$, with probability at least $1 - \delta$ over the samples one can construct an algorithm that returns a solution $S \subseteq N$ s.t,

$$f(S) - \min_{T \subseteq N} f(T) \leq \alpha.$$

This framework was recently introduced in [BRS17] for the problem of submodular maximization where the standard notion of approximation is multiplicative. For submodular minimization, since the optimum may have zero value, the suitable measure is that of additive approximations for $[0,1]$-bounded functions, and the goal is to obtain a solution which is a $o(1)$ additive approximation to the minimum (see e.g. [CLSW16, EN15, SK10]). The question is then:

*Can submodular functions be minimized from samples?*

Since submodular functions can be minimized in polynomial-time, it is tempting to conjecture that when the function is learnable it also has desirable approximation guarantees from samples, especially in light of positive results in related settings of *submodular maximization*:

- **Constrained maximization.** For functions that can be maximized in polynomial time under a cardinality constraint, like modular and unit-demand functions, there are polynomial time algorithms that obtain an arbitrarily good approximation using polynomially-many samples [BRS16, BRS17]. For general monotone submodular functions which are NP-hard to maximize under cardinality constraints, there is no algorithm that can obtain a reasonable approximation from polynomially-many samples [BRS17]. For the problem of unconstrained minimization, submodular functions can be optimized in polynomial time;

- **Unconstrained maximization.** For unconstrained maximization of general submodular functions, the problem is NP-hard to maximize (e.g. `MAX-CUT`) and one seeks constant factor approximations. For this problem, there is an extremely simple algorithm that uses no queries and obtains a good approximation: choose elements uniformly at random with probability $1/2$ each. This algorithm achieves a constant factor approximation of $1/4$ for general submodular functions. For *symmetric* submodular functions (i.e. $f(S) = f(N \setminus S)$), this algorithm is a $1/2$-approximation which is optimal, since no algorithm can obtain an approximation ratio strictly better than $1/2$ using polynomially-many *value* queries, even for symmetric submodular functions [FMV11]. For unconstrained symmetric submodular *minimization*, there is an appealing analogue: the empty set and the ground set $N$ are guaranteed to be minimizers of the function (see Section 2). This algorithm, of course, uses no queries either. The parallel between these two problems seems quite intuitive, and it is tempting to conjecture that like for unconstrained submodular maximization, there are optimization from samples algorithms for general unconstrained submodular minimization with good approximation guarantees.

**Main result.** Somewhat counter-intuitively, we show that despite being computationally tractable to optimize, submodular functions cannot be minimized from samples to within a desirable guarantee, even when these functions are learnable. In particular, we show that there is no algorithm for minimizing a submodular function from polynomially-many samples drawn from any distribution that obtains an additive approximation of $1/2 - o(1)$, even when the function is `PAC`-learnable. Furthermore, we show that this bound is tight: the algorithm which returns the empty set or ground set each with probability $1/2$ achieves at least a $1/2$ approximation. Notice that this also implies that in general, there is no learning algorithm that can produce a surrogate whose minima is close to the minima of the function we aim to optimize, as otherwise this would contradict our main result.

**Technical overview.** At a high level, hardness results in optimization from samples are shown by constructing a family of functions, where the values of functions in the family are likely to be indistinguishable for the samples, while having very different optimizers. The main technical difficulty is to construct a family of functions that concurrently satisfy these two properties (indistinguishability and different optimizers), and that are also `PAC`-learnable. En route to our main construction, we first construct a family of functions that are completely indistinguishable given samples drawn from the uniform distribution, in which case we obtain a $1/2 - o(1)$ impossibility result (Section 2). The

general result that holds for any distribution requires heavier machinery to argue about more general families of functions where some subset of functions can be distinguished from others given samples. Instead of satisfying the two desired properties for all functions in a fixed family, we show that these properties hold for all functions in a randomized subfamily (Section 3.2). We then develop an efficient learning algorithm for the family of functions constructed for the main hardness result (Section 3.3). This algorithm builds multiple linear regression predictors and a classifier to direct a fresh set to the appropriate linear predictor. The learning of the classifier and the linear predictors relies on multiple observations about the specific structure of this class of functions.

## 1.1 Related work

The problem of optimization from samples was introduced in the context of constrained submodular maximization [BRS17, BRS16]. In general, for maximizing a submodular function under a cardinality constraint, no algorithm can obtain a constant factor approximation guarantee from any samples. As discussed above, for special classes of submodular functions that can be optimized in polynomial time under a cardinality constraint, and for unconstrained maximization, there are desirable optimization from samples guarantees. It is thus somewhat surprising that submodular minimization, which is an unconstrained optimization problem that is optimizable in polynomial time in the value query model, is hard to optimize from samples. From a technical perspective the constructions are quite different. In maximization, the functions constructed in [BRS17, BRS16] are monotone so the ground set would be an optimal solution if the problem was unconstrained. Instead, we need to construct novel non-monotone functions. In convex optimization, recent work shows a tight $1/2$-inapproximability for convex minimization from samples [BS17]. Although there is a conceptual connection between that paper and this one, from a technical perspective these papers are orthogonal. The discrete analogue of the family of convex functions constructed in that paper is not (even approximately) a family of submodular functions, and the constructions are significantly different.

# 2 Warm up: the Uniform Distribution

As a warm up to our main impossibility result, we sketch a tight lower bound for the special case in which the samples are drawn from the uniform distribution. At a high level, the idea is to construct a function which considers some special subset of "good" elements that make its value drops when a set contains *all* such "good" elements. When samples are drawn from the uniform distribution and "good" elements are sufficiently rare, there is a relatively simple construction that obfuscates which elements the function considers "good", which then leads to the inapproximability.

## 2.1 Hardness for uniform distribution

We construct a family of functions $\mathcal{F}$ where $f_i \in \mathcal{F}$ is defined in terms of a set $G_i \subset N$ of size $\sqrt{n}$. For each such function we call $G_i$ the set of *good* elements, and $B_i = N \setminus G_i$ its *bad* elements. We denote the number of good and bad elements in a set $S$ by $g_S$ and $b_S$, dropping the subscripts ($S$ and $i$) when clear from context, so $g = |G_i \cap S|$ and $b = |B_i \cap S|$. The function $f_i$ is defined as follows:

$$f_i(S) := \begin{cases} \dfrac{1}{2} + \dfrac{1}{2n} \cdot (g + b) & \text{if } g < \sqrt{n} \\ \dfrac{1}{2n} \cdot b & \text{if } g = \sqrt{n} \end{cases}$$

It is easy to verify that these functions are submodular with range in $[0, 1]$ (see illustration in Figure 1a). Given samples drawn uniformly at random (u.a.r.), it is impossible to distinguish good and bad elements since with high probability (w.h.p.) $g < \sqrt{n}$ for all samples. Informally, this implies that a good learner for $\mathcal{F}$ over the uniform distribution $\mathcal{D}$ is $f'(S) = 1/2 + |S|/(2n)$.

Intuitively, $\mathcal{F}$ is not $1/2 - o(1)$ optimizable from samples because if an algorithm cannot learn the set of good elements $G_i$, then it cannot find $S$ such $f_i(S) < 1/2 - o(1)$ whereas the optimal solution $S_i^\star = G_i$ has value $f_i(G_i) = 0$.

**Theorem 1.** *Submodular functions $f : 2^N \to [0, 1]$ are not $1/2 - o(1)$ optimizable from samples drawn from the uniform distribution for the problem of submodular minimization.*

*Proof.* The details for the derivation of concentration bounds are in Appendix B. Consider $f_k$ drawn u.a.r. from $\mathcal{F}$ and let $f^\star = f_k$ and $G^\star = G_k$. Since the samples are all drawn from the uniform distribution, by standard application of the Chernoff bound we have that every set $S_i$ in the sample respects $|S_i| \leq 3n/4$, w.p. $1 - e^{-\Omega(n)}$. For sets $S_1, \ldots, S_m$, all of size at most $3n/4$, when $f_j$ is drawn u.a.r. from $\mathcal{F}$ we get that $|S_i \cap G_j| < \sqrt{n}$, w.p. $1 - e^{-\Omega(n^{1/2})}$ for all $i \in [m]$, again by Chernoff, and since $m = \mathrm{poly}(n)$. Notice that this implies that w.p. $1 - e^{-\Omega(n^{1/2})}$ for all $i \in [m]$:

$$f_j(S_i) = \frac{1}{2} + \frac{|S_i|}{2n}$$

Now, let $\mathcal{F}'$ be the collection of all functions $f_j$ for which $f_j(S_i) = 1/2 + |S_i|/(2n)$ on all sets $\{S_i\}_{i=1}^m$. The argument above implies that $|\mathcal{F}'| = (1 - e^{-\Omega(n^{1/2})})|\mathcal{F}|$. Thus, since $f^\star$ is drawn u.a.r. from $\mathcal{F}$ we have that $f^\star \in \mathcal{F}'$ w.p. $1 - e^{-\Omega(n^{1/2})}$, and we condition on this event.

Let $S$ be the (possibly randomized) solution returned by the algorithm. Observe that $S$ is *independent* of $f^\star \in \mathcal{F}'$. In other words, the algorithm cannot learn any information about which function in $\mathcal{F}'$ generates the samples. By Chernoff, if we fix $S$ and choose $f$ u.a.r. from $\mathcal{F}$, then, w.p. $1 - e^{-\Omega(n^{1/6})}$:

$$f(S) \geq \frac{1}{2} - o(1).$$

Since $\frac{|\mathcal{F}'|}{|\mathcal{F}|} = 1 - e^{-\Omega(n^{1/2})}$, it is also the case that $f^\star(S) \geq 1/2 - o(1)$ w.p. $1 - e^{-\Omega(n^{1/6})}$ over the choice of $f^\star \in \mathcal{F}'$. By the probabilistic method and since all the events we conditioned on occur with exponentially high probability, there exists $f^\star \in \mathcal{F}$ s.t. the value of the set $S$ returned by the algorithm is $1/2 - o(1)$ whereas the optimal solution is $f^\star(G^\star) = 0$. □

## 2.2 A tight upper bound

We now show that the result above is tight. In particular, by randomizing between the empty set and the ground set we get a solution whose value is at most $1/2$. In the case of symmetric functions, i.e. $f(S) = f(N \setminus S)$ for all $S \subseteq N$, $\emptyset$ and $N$ are minima since $f(N) + f(\emptyset) \leq f(S) + f(N \setminus S)$ for all $S \subseteq N$ as shown below.[1] Notice, that this does not require any samples.

**Proposition 2.** *The algorithm which returns the empty set $\emptyset$ or the ground $N$ with probability $1/2$ each is a $1/2$ additive approximation for the problem of unconstrained submodular minimization.*

*Proof.* Let $S \subseteq N$, observe that

$$f(N \setminus S) - f(\emptyset) \geq f((N \setminus S) \cup S) - f(S) = f(N) - f(S)$$

where the inequality is by submodularity. Thus, we obtain

$$\frac{1}{2}(f(N) + f(\emptyset)) \leq \frac{1}{2}f(S) + \frac{1}{2}f(N \setminus S) \leq f(S) + \frac{1}{2}.$$

In particular, this holds for $S \in \mathrm{argmin}_{T \subseteq N} f(T)$. □

## 3 General Distribution

In this section, we show our main result, namely that there exists a family of submodular functions such that, despite being `PAC`-learnable for all distributions, no algorithm can obtain an approximation better than $1/2 - o(1)$ for the problem of unconstrained minimization.

The functions in this section build upon the previous construction, though are inevitably more involved in order to achieve learnability and inapproximability on *any* distribution. The functions constructed for the uniform distribution do not yield inapproximability for general distributions due to the fact that the indistinguishability between two functions no longer holds when sets $S$ of large size are sampled with non-negligible probability. Intuitively, in the previous construction, once a set is sufficiently large the good elements of the function can be distinguished from the bad ones. The main idea to get around this issue is to introduce *masking* elements $M$. We construct functions such that, for sets $S$ of large size, good and bad elements are indistinguishable if $S$ contains *at least one* masking element.

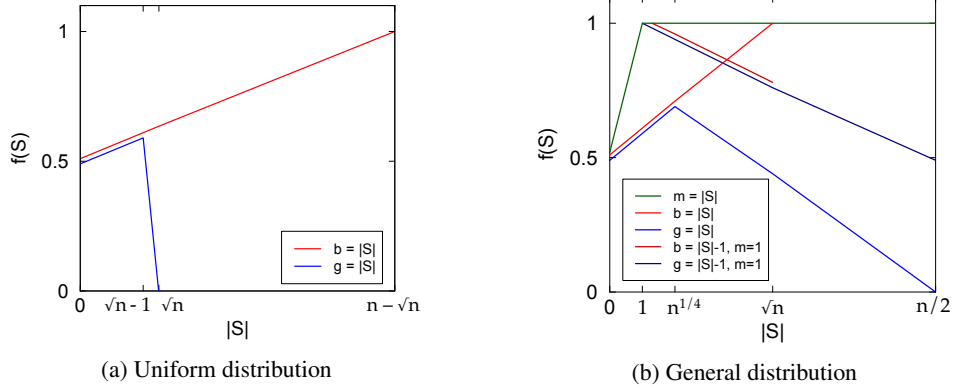

Figure 1: An illustration of the value of a set $S$ of good (blue), bad (red), and masking (green) elements as a function of $|S|$ for the functions constructed. For the general distribution case, we also illustrate the value of a set $S$ of good (dark blue) and bad (dark red) elements when $S$ also contains at least one masking element.

**The construction.** Each function $f_i \in \mathcal{F}$ is defined in terms of a partition $P_i$ of the ground set into *good*, *bad*, and *masking* elements. The partitions we consider are $P_i = (G_i, B_i, M_i)$ with $|G_i| = n/2$, $|B_i| = \sqrt{n}$, and $|M_i| = n/2 - \sqrt{n}$. Again, when clear from context, we drop indices of $i$ and $S$ and the number of good, bad, and masking elements in a set $S$ are denoted by $g, b$, and $m$. For such a given partition $P_i$, the function $f_i$ is defined as follows (see illustration in Figure 1b):

$$f_i(S) = \frac{1}{2} + \begin{cases} \frac{1}{2\sqrt{n}} \cdot (b+g) & \textbf{Region } \mathcal{X}: \text{ if } m = 0 \text{ and } g < n^{1/4} \\ \frac{1}{2\sqrt{n}} \cdot \left(b + n^{1/4}\right) - \frac{1}{n} \cdot \left(g - n^{1/4}\right) & \textbf{Region } \mathcal{Y}: \text{ if } m = 0 \text{ and } g \geq n^{1/4} \\ \frac{1}{2} - \frac{1}{n} \cdot (b+g) & \textbf{Region } \mathcal{Z}: \text{ otherwise} \end{cases}$$

### 3.1 Submodularity

In the appendix, we prove that the functions $f_i$ constructed as above are indeed submodular (Lemma 10). By rescaling $f_i$ with an additive term of $n^{1/4}/(2\sqrt{n}) = 1/(2n^{1/4})$, it can be easily verified that its range is in $[0, 1]$. We use the non-normalized definition as above for ease of notation.

### 3.2 Inapproximability

We now show that $\mathcal{F}$ cannot be minimized within a $1/2 - o(1)$ approximation given samples from any distribution. We first define $\mathcal{F}^M$, which is a randomized subfamily of $\mathcal{F}$. We then give a general lemma that shows that if two conditions of indistinguishability and gap are satisfied then we obtain inapproximability. We then show that these two conditions are satisfied for the subfamily $\mathcal{F}^M$.

**A randomization over masking elements.** Instead of considering a function $f$ drawn u.a.r. from $\mathcal{F}$ as in the uniform case, we consider functions $f$ in a *randomized subfamily* of functions $\mathcal{F}^M \subseteq \mathcal{F}$ to obtain the indistinguishability and gap conditions. Given the family of functions $\mathcal{F}$, let $M$ be a *uniformly random subset* of size $n/2 - \sqrt{n}$ and define $\mathcal{F}^M \subset \mathcal{F}$:

$$\mathcal{F}^M := \{f_i \in \mathcal{F} : (G_i, B_i, M)\}.$$

Since masking elements are distinguishable from good and bad elements, they need to be the same set of elements for each function in family $\mathcal{F}^M$ to obtain indistinguishability of functions in $\mathcal{F}^M$.

**The inapproximability lemma.** In addition to this randomized subfamily of functions, another main conceptual departure of the following inapproximability lemma from the uniform case is that no assumption can be made about the samples, such as their size, since the distribution is arbitrary. We denote by $U(A)$ the uniform distribution over the set $A$.

**Lemma 3.** *Let $\mathcal{F}$ be a family of functions and $\mathcal{F}' = \{f_1, \ldots, f_p\} \subseteq \mathcal{F}$ be a subfamily of functions drawn from some distribution. Assume the following two conditions hold:*

1. ***Indistinguishability.** For all $S \subseteq N$, w.p. $1 - e^{-\Omega(n^{1/4})}$ over $\mathcal{F}'$: for every $f_i, f_j \in \mathcal{F}'$,*
$$f_i(S) = f_j(S);$$

2. ***$\alpha$-gap.** Let $S_i^\star$ be a minimizer of $f_i$, then w.p. $1$ over $\mathcal{F}'$: for all $S \subseteq N$,*
$$\mathbb{E}_{f_i \sim U(\mathcal{F}')}[f_i(S) - f_i(S_i^\star)] \geq \alpha;$$

*Then, $\mathcal{F}$ is not $\alpha$-minimizable from strictly less than $e^{\Omega(n^{1/4})}$ samples over any distribution $\mathcal{D}$.*

Note that the ordering of the quantifiers is crucial. The proof is deferred to the appendix, but the main ideas are summarized as follows. We use a probabilistic argument to switch from the randomization over $\mathcal{F}'$ to the randomization over $S \sim \mathcal{D}$ and show that there exists a deterministic $F \subseteq \mathcal{F}$ such that $f_i(S) = f_j(S)$ for all $f_i, f_j \in F$ w.h.p. over $S \sim \mathcal{D}$. By a union bound this holds for all samples $S$. Thus, for such a family of functions $F = \{f_1, \ldots, f_p\}$, the choices of an algorithm that is given samples from $f_i$ for $i \in [p]$ are *independent* of $i$. By the $\alpha$-gap condition, this implies that there exists $f_i \in F$ for which a solution $S$ returned by the algorithm is at least $\alpha$ away from $f_i(S_i^\star)$.

**Indistinguishability and gap of $\mathcal{F}$.** We now show the indistinguishability and gap conditions, with $\alpha = 1/2 - o(1)$, which immediately imply a $1/2 - o(1)$ inapproximability by Lemma 3. For the indistinguishability, it suffices to show that good and bad elements are indistinguishable since the masking elements are identical for all functions in $\mathcal{F}^M$. Good and bad elements are indistinguishable since, w.h.p., a set $S$ is not in region $\mathcal{Y}$, which is the only region distinguishing good and bad elements.

**Lemma 4.** *For all $S \subseteq N$ s.t. $|S| < n^{1/4}$: For all $f_i \in \mathcal{F}^M$,*
$$f_i(S) = \frac{1}{2} + \begin{cases} \frac{1}{2\sqrt{n}} \cdot (b+g) & \text{if } m = 0 \text{ (Region } \mathcal{X}) \\ \frac{1}{2} - \frac{1}{n} \cdot (b+g) & \text{otherwise (Region } \mathcal{Z}) \end{cases}$$

*and for all $S \subseteq N$ such that $|S| \geq n^{1/4}$, with probability $1 - e^{-\Omega(n^{1/4})}$ over $\mathcal{F}^M$: For all $f_i \in \mathcal{F}^M$,*
$$f_i(S) = 1 - \frac{1}{n} \cdot (b+g) \qquad \text{(Region } \mathcal{Z}).$$

*Proof.* Let $S \subseteq N$. If $|S| < n^{1/4}$, then the proof follows immediately from the definition of $f_i$. If $|S| \geq n^{1/4}$, then, the number of masking elements $m$ in $S$ is $m = |M \cap S|$ for all $f_i \in \mathcal{F}^M$. We then get $m \geq 1$, for all $f_i \in \mathcal{F}^M$, with probability $1 - e^{-\Omega(n^{1/4})}$ over $\mathcal{F}^M$ by Chernoff bound. The proof then follows again immediately from the definition of $f_i$. $\qquad \square$

Next, we show the gap. The gap is since the good elements can be any subset of $N \setminus M$.

**Lemma 5.** *Let $S_i^\star$ be a minimizer of $f_i$. With probability $1$ over $\mathcal{F}^M$, for all $S \subseteq N$,*
$$\mathbb{E}_{f_i \sim U(\mathcal{F}^M)}[f_i(S)] \geq \frac{1}{2} - o(1).$$

*Proof.* Let $S \subseteq N$ and $f_i \sim U(\mathcal{F}^M)$. Note that the order of the quantifiers in the statement of the lemma implies that $S$ can be dependent on $M$, but that it is independent of $i$. There are three cases. If $m \geq 1$, then $S$ is in region $\mathcal{Z}$ and $f_i(S) \geq 1/2$. If $m = 0$ and $|S| \leq n^{7/8}$, then $S$ is in region $\mathcal{X}$ or $\mathcal{Y}$ and $f_i(S) \geq 1/2 - n^{7/8}/n = \frac{1}{2} - o(1)$. Otherwise, $m = 0$ and $|S| \geq n^{7/8}$. Since $S$ is independent of $i$, by Chernoff bound, we get

$$(1 - o(1)) \cdot |S| \leq \frac{n/2 + \sqrt{n}}{\sqrt{n}} \cdot b \quad \text{and} \quad \frac{n/2 + \sqrt{n}}{n/2} \cdot g \leq (1 + o(1)) \cdot |S|$$

with probability $1 - e^{-\Omega(n^{1/4})}$. Thus $S$ is in region $\mathcal{Y}$ and

$$f_i(S) \geq \frac{1}{2} + (1 - o(1)) \frac{1}{2\sqrt{n}} \cdot \frac{\sqrt{n}}{n/2 + \sqrt{n}} \cdot |S| - (1 + o(1)) \frac{1}{n} \cdot \frac{n/2}{n/2 + \sqrt{n}} \cdot |S| \geq \frac{1}{2} - o(1).$$

Thus, we obtain $\mathbb{E}_{f_i \sim U(\mathcal{F}^M)}[f_i(S)] \geq \frac{1}{2} - o(1)$. $\qquad \square$

Combining the above three lemmas, we obtain the inapproximability result.

**Lemma 6.** *The problem of submodular minimization cannot be approximated with a $1/2 - o(1)$ additive approximation given $\mathrm{poly}(n)$ samples from any distribution $\mathcal{D}$.*

*Proof.* For any set $S \subseteq N$, observe that the number $g + b$ of elements in $S$ that are either good or bad is the same for any two functions $f_i, f_j \in \mathcal{F}^M$ and for any $\mathcal{F}^M$. Thus, by Lemma 4, we obtain the indistinguishability condition. Next, the optimal solution $S_i^\star = G_i$ of $f_i$ has value $f_i(G_i) = o(1)$, so by Lemma 5, we obtain the $\alpha$-gap condition with $\alpha = 1/2 - o(1)$. Thus $\mathcal{F}$ is not $1/2 - o(1)$ minimizable from samples from any distribution $\mathcal{D}$ by Lemma 3. The class of functions $\mathcal{F}$ is a class of submodular functions by Lemma 10 (in Appendix C). □

## 3.3 Learnability of $\mathcal{F}$

We now show that every function in $\mathcal{F}$ is efficiently learnable from samples drawn from any distribution $\mathcal{D}$. Specifically, we show that for any $\epsilon, \delta \in (0, 1)$ the functions are $(\epsilon, \delta) - \mathtt{PAC}$ learnable with the absolute loss function (or any Lipschitz loss function) using $\mathrm{poly}(1/\epsilon, 1/\delta, n)$ samples and running time. At a high level, since each function $f_i$ is piecewise-linear over three different regions $\mathcal{X}_i, \mathcal{Y}_i$, and $\mathcal{Z}_i$, the main idea is to exploit this structure by first training a classifier to distinguish between regions and then apply linear regression in different regions.

**The learning algorithm.** Since every function $f \in \mathcal{F}$ is piecewise linear over three different regions, there are three different linear functions $f_\mathcal{X}, f_\mathcal{Y}, f_\mathcal{Z}$ s.t. for every $S \subseteq N$ its value $f(S)$ can be expressed as $f_\mathcal{R}(S)$ for some region $\mathcal{R} \in \{\mathcal{X}, \mathcal{Y}, \mathcal{Z}\}$. The learning algorithm produces a predictor $\tilde{f}$ by using a multi-label classifier and a set of linear predictors $\{f_{\tilde{\mathcal{X}}}, f_{\tilde{\mathcal{Y}}}\} \cup \{\cup_{i \in \tilde{M}} f_{\tilde{\mathcal{Z}}_i}\}$. The multi-label classifier creates a mapping from sets to regions, $g : 2^N \to \{\tilde{\mathcal{X}}, \tilde{\mathcal{Y}}\} \cup \{\cup_{i \in \tilde{M}} \tilde{\mathcal{Z}}_i\}$, s.t. $\mathcal{X}, \mathcal{Y}, \mathcal{Z}$ are approximated by $\tilde{\mathcal{X}}, \tilde{\mathcal{Y}}, \cup_{i \in \tilde{M}} \tilde{\mathcal{Z}}_i$. Given a sample $S \sim \mathcal{D}$, using the algorithm then retuns $\tilde{f}(S) = f_{g(S)}(S)$. We give a formal description below (detailed description is in Appendix D).

---

**Algorithm 1** A learning algorithm for $f \in \mathcal{F}$ which combines classification and linear regression.

---

**Input:** samples $\mathcal{S} = \{(S_j, f(S_j))\}_{j \in [m]}$

$(\tilde{\mathcal{Z}}, \tilde{M}) \leftarrow (\emptyset, \emptyset)$
**for** $i = 1$ **to** $n$ **do**
$\quad \tilde{\mathcal{Z}}_i \leftarrow \{S : a_i \in S, S \notin \tilde{\mathcal{Z}}\}$
$\quad f_{\tilde{\mathcal{Z}}_i} \leftarrow \mathrm{ERM}^{\mathrm{reg}}(\{(S_j, f(S_j)) : S_j \in \tilde{\mathcal{Z}}_i\})$  \hfill linear regression
$\quad$ **if** $\sum_{(S_j, f(S_j)) : S_j \in \tilde{\mathcal{Z}}_i} |f_{\tilde{\mathcal{Z}}_i}(S_j) - f(S_j)| = 0$ **then**
$\quad\quad \tilde{\mathcal{Z}} \leftarrow \tilde{\mathcal{Z}} \cup \tilde{\mathcal{Z}}_i, \tilde{M} \leftarrow \tilde{M} \cup \{a_i\}$
$C \leftarrow \mathrm{ERM}^{\mathrm{cla}}(\{(S_j, f(S_j)) : S_j \notin \tilde{\mathcal{Z}}, j \leq m/2\})$  \hfill train a classifier for regions $\mathcal{X}, \mathcal{Y}$
$(\tilde{\mathcal{X}}, \tilde{\mathcal{Y}}) \leftarrow (\{S : S \notin \tilde{\mathcal{Z}}, C(S) = 1\}, \{S : S \notin \tilde{\mathcal{Z}}, C(S) = -1\})$
**return** $\tilde{f} \leftarrow S \mapsto \begin{cases} |S|/(2\sqrt{n}) & \text{if } S \in \tilde{\mathcal{X}} \\ f_{\tilde{\mathcal{Y}}}(S) = \mathrm{ERM}^{\mathrm{reg}}(\{(S_j, f(S_j)) : S_j \in \tilde{\mathcal{Y}}, j > m/2\}) & \text{if } S \in \tilde{\mathcal{Y}} \\ f_{\tilde{\mathcal{Z}}_i}(S) : i = \min(\{i' : a_{i'} \in S \cap \tilde{M}\}) & \text{if } S \in \tilde{\mathcal{Z}} \end{cases}$

---

**Overview of analysis of the learning algorithm.** There are two main challenges in training the algorithm. The first is that the region $\mathcal{X}, \mathcal{Y}$, or $\mathcal{Z}$ that a sample $(S_j, f(S_j))$ belongs to is not known. Thus, even before being able to train a classifier which learns the regions $\tilde{\mathcal{X}}, \tilde{\mathcal{Y}}, \tilde{\mathcal{Z}}$ using the samples, we need to learn the region a sample $S_j$ belongs to using $f(S_j)$. The second is that the samples $\mathcal{S}_\mathcal{R}$ used for training a linear regression predictor $f_\mathcal{R}$ over region $\mathcal{R}$ need to be carefully selected so that $\mathcal{S}_\mathcal{R}$ is a collection of i.i.d. samples from the distribution $S \sim \mathcal{D}$ conditioned on $S \in \mathcal{R}$ (Lemma 20).

We first discuss the challenge of labeling samples with the region they belong to. Observe that for a fixed masking element $a_i \in M$, $f \in \mathcal{F}$ is linear over all sets $S$ containing $a_i$ since these sets are all in region $\mathcal{Z}$. Thus, there must exist a linear regression predictor $f_{\tilde{\mathcal{Z}}_i} = \mathrm{ERM}^{\mathrm{reg}}(\cdot)$ with zero empirical loss over all samples $S_j$ containing $a_i$ if $a_i \in M$ (and thus $S_j \in \mathcal{Z}$). $\mathrm{ERM}^{\mathrm{reg}}(\cdot)$ minimizes the empirical loss on the input samples over the class of linear regression predictors with bounded norm

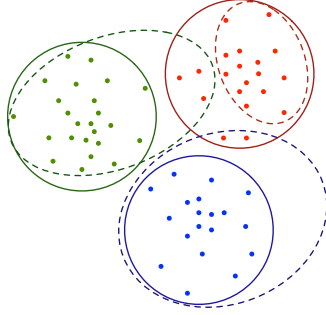

Figure 2: An illustration of the regions. The dots represent the samples, the corresponding full circles represent the regions $\mathcal{X}$ (red), $\mathcal{Y}$ (blue), and $\mathcal{Z}$ (green). The ellipsoids represent the regions $\tilde{\mathcal{X}}, \tilde{\mathcal{Y}}, \tilde{\mathcal{Z}}$ learned by the classifier. Notice that $\tilde{\mathcal{Z}}$ has no false negatives.

(Lemma 19). If $f_{\tilde{\mathcal{Z}}_i}$ has zero empirical loss, we directly classify any set $S$ containing $a_i$ as being in $\tilde{\mathcal{Z}}$. Next, for a sample $(S_j, f(S_j))$ not in $\tilde{\mathcal{Z}}$, we can label these samples since $S_j \in \mathcal{X}$ if and only if $f(S_j) = |S_j|/(2\sqrt{n})$. With these labeled samples $\mathcal{S}'$, we train a binary classifier $C = \text{ERM}^{\text{cla}}(\mathcal{S}')$ that indicates if $S$ s.t. $S \notin \tilde{\mathcal{Z}}$ is in region $\tilde{\mathcal{X}}$ or $\tilde{\mathcal{Y}}$. $\text{ERM}^{\text{cla}}(\mathcal{S}')$ minimizes the empirical loss on labeled samples $\mathcal{S}'$ over the class of halfspaces $\mathbf{w} \in \mathbb{R}^n$ (Lemma 23).

Regarding the second challenge, we cannot use all samples $S_j$ s.t. $S_j \in \tilde{\mathcal{Y}}$ to train a linear predictor $f_{\tilde{\mathcal{Y}}}$ for region $\tilde{\mathcal{Y}}$ since these same samples were used to define $\tilde{\mathcal{Y}}$, so they are not a collection of i.i.d. samples from the distribution $S \sim \mathcal{D}$ conditioned on $S \in \tilde{\mathcal{Y}}$. To get around this issue, we partition the samples into two distinct collections, one to train the classifier $C$ and one to train $f_{\tilde{\mathcal{Y}}}$ (Lemma 24). Next, given $T \in \tilde{\mathcal{Z}}$, we predict $f_{\tilde{\mathcal{Z}}_i}(T)$ where $i$ is s.t. $a_i \in T \cap \tilde{M}$ (breaking ties lexicographically) which performs well since $\tilde{f}_{\tilde{\mathcal{Z}}_i}$ has zero empirical error for $a_i \in \tilde{M}$ (Lemma 22). Since we break ties lexicographically, $\tilde{f}_{\tilde{\mathcal{Z}}_i}$ must be trained over samples $S_j$ such that $a_i \in S_j$ and $a_{i'} \notin S_j$ for $i'$ s.t. $i' < i$ and $a_{i'} \in \tilde{M}$ to obtain i.i.d. samples from the same distribution as $T \sim \mathcal{D}$ conditioned on $T$ being directed to $\tilde{f}_{\tilde{\mathcal{Z}}_i}$ (Lemma 21).

The analysis of the learning algorithm leads to the following main learning result.

**Lemma 7.** *Let $\tilde{f}$ be the predictor returned by Algorithm 1, then w.p. $1 - \delta$ over $m \in O(n^3 + n^2(\log(2n/\delta))/\epsilon^2)$ samples $\mathcal{S}$ drawn i.i.d. from any distribution $\mathcal{D}$, $\mathbb{E}_{S \sim \mathcal{D}}[|\tilde{f}(S) - f(S)|] \le \epsilon$.*

### 3.4 Main Result

We conclude this section with our main result which combines Lemmas 6 and 7.

**Theorem 8.** *There exists a family of $[0, 1]$-bounded submodular functions $\mathcal{F}$ that is efficiently* PAC*-learnable and that cannot be optimized from polynomially many samples drawn from any distribution $\mathcal{D}$ within a $1/2 - o(1)$ additive approximation for unconstrained submodular minimization.*

## 4 Discussion

In this paper, we studied the problem of submodular minimization from samples. Our main result is an impossibility, showing that even for learnable submodular functions it is impossible to find a non-trivial approximation to the minimizer with polynomially-many samples, drawn from any distribution. In particular, this implies that minimizing a general submodular function learned from data cannot yield desirable guarantees. In general, it seems that the intersection between learning and optimization is elusive, and a great deal still remains to be explored.

## Footnotes

[1] Although $\emptyset$ and $N$ are trivial minima if $f$ is symmetric, the problem of minimizing a symmetric submodular function over proper nonempty subsets is non-trivial (see [Que98]).

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
