[Supplementary Material]

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

## A  Optimization from Samples via Learning then Optimization

In this section, we discuss the issues with the approach which consists of first learning a surrogate of the true function and then optimizing this surrogate. A first issue is that the surrogate may be non-submodular, and even though we might be able to approximate the true function everywhere, the surrogate may be intractable to optimize (Section A.1). A second issue is that the surrogate is submodular and approximates the true function on all the samples but that its minimum is far from the minimum of the true function (Section A.2).

### A.1  The surrogate is a good approximation of the true function but is not submodular

Consider the following submodular function defined over a partition of the ground set $N$ into a set of good elements $G$ and a set of bad elements $B$, each of size $n/2$:

$$f(S) = \frac{1}{2} + \frac{1}{n} \left( |S \cap B| - |S \cap G| \right).$$

Let $\tilde{f}$ be a surrogate for $f$ defined as follows:

$$\tilde{f}(S) = \frac{1}{2} + \begin{cases} \frac{1}{n} \left( |S \cap B| - |S \cap G| \right) & \text{if } ||S \cap B| - |S \cap G|| \geq \frac{1}{2}\epsilon n \\ 0 & \text{otherwise} \end{cases}$$

It is easy to verify that the surrogate $\tilde{f}$ $\epsilon$-approximates $f$ on all sets. However $\tilde{f}$ is intractable to optimize. Intuitively, by concentration bounds, a query $S$ of the algorithm has value $\tilde{f}(S) = 1/2$ for an exponentially high fraction of the sets $S$. Thus, with polynomially many adaptive queries $(S, f(S))$ of the choice of the algorithm, the algorithm will probably not be able to distinguish $\tilde{f}(S)$ from the constant function $1/2$ everywhere. Since the optimal solution is $S^\star = G$ and has value $0$, no algorithm can then do better than the trivial $1/2$-approximation to minimize $\tilde{f}$.

### A.2  The surrogate is submodular but its minimum is far from the true minimum

We illustrate the second issue with samples from the uniform distribution. Similarly, consider the following submodular function defined over a partition of the ground set $N$ into a set of good elements $G$ and a set of bad elements $B$, each of size $n/2$

$$f(S) := \begin{cases} \frac{1}{2} + \frac{1}{2n} \cdot \left( |S \cap G| + |S \cap B| \right) & \text{if } |S \cap G| < n/2 \\ \frac{1}{2n} \cdot |S \cap B| & \text{if } |S \cap G| = n/2 \end{cases}$$

This function is similar to the functions constructed in Section 2. Let $\tilde{f}$ be a surrogate for $f$ where $G$ and $B$ are interchanged:

$$\tilde{f}(S) := \begin{cases} \frac{1}{2} + \frac{1}{2n} \cdot \left( |S \cap B| + |S \cap G| \right) & \text{if } |S \cap B| < n/2 \\ \frac{1}{2n} \cdot |S \cap G| & \text{if } |S \cap B| = n/2 \end{cases}$$

It is easy to verify that given polynomially many samples $S$ from the uniform distribution, $\tilde{f}(S) = f(S)$ with high probability, so $\tilde{f}$ is consistent with all the samples. However its minimum $B$, which is such that $\tilde{f}(B) = 0$ is a bad solution for the true underlying function since $f(B) = 3/4$.

## B  Concentration bounds

**Lemma 9** (Chernoff Bound). *Let $X_1, \ldots, X_n$ be independent indicator random variables. Let $X = \sum_{i=1}^{n} X_i$ and $\mu = \mathbb{E}[X]$. For $0 < \delta < 1$,*

$$\Pr[|X - \mu| \geq \delta\mu] \leq 2e^{-\mu\delta^2/3}.$$

## B.1 Concentration bounds from Section 2

- Every set $S_i$ in the sample respects $|S_i| \leq 3n/4$, w.p. $1 - e^{-\Omega(n)}$. Consider a set $S_i \sim U$. Let $X_j$ be an indicator variable indicating if $j \in S_i$. By Chernoff bound with $|S| = \sum_{j=1}^{n} X_j$, $\mu = n/2$, $\delta = 1/2$, $\Pr[|S_i| - n/2 \geq n/4] \leq 2e^{-n/24}$. By a union bound over all polynomially many samples, the claim holds with probability $1 - e^{-\Omega(n)}$.

- For sets $S_1, \ldots, S_m$, $m = \text{poly}(n)$, all of size at most $3n/4$, when $f_j$ is drawn u.a.r. from $\mathcal{F}$ we get that $|S_i \cap G_j| < \sqrt{n}$, w.p. $1 - e^{-\Omega(n^{1/2})}$. Consider a set $T$ of size $n/4$. Let $X_1, \ldots, X_{n/4}$ be indicator variables indicating if elements in $T$ are also in $G_j$. By Chernoff bound with $|T \cap G_j| = \sum_{i=1}^{n/4} X_i$, $\mu = n/4 \cdot \sqrt{n}/n = \sqrt{n}/4$, $\delta = 1/2$, $\Pr[|T \cap G_j| - \sqrt{n}/4 \geq \sqrt{n}/8] \leq 2e^{-n^{1/2}/48}$. Thus, with probability $1 - e^{-\Omega(n^{1/2})}$, $|T \cap G_j| > 0$, and for a set $S$ of size at most $3n/4$, $|(N \setminus S) \cap G_j| > 0$ and $G_j \not\subseteq S$. Thus, by a union bound, it holds with probability $1 - e^{-\Omega(n^{1/2})}$ over $f_j$ that for all samples $S$, $G_j \not\subseteq S$.

- Fix $S$ and choose $f$ u.a.r. from $\mathcal{F}$, then, w.p. $1 - e^{-\Omega(n^{1/6})}$:

$$f(S) \geq \frac{1}{2} - o(1).$$

  If $|S| \leq n - n^{2/3}$, let $T \subseteq N \setminus S$ such that $|T| = n^{2/3}$. Let $X_1, \ldots, X_{n^{2/3}}$ be indicator variables indicating if elements in $T$ are also in $G$. By Chernoff bound with $|T \cap G| = \sum_{i=1}^{n^{2/3}} X_i$, $\mu = n^{2/3} \cdot \sqrt{n}/n = n^{1/6}$, $\delta = 1/2$, $\Pr[|T \cap G| - n^{1/6} \geq n^{1/6}/2] \leq 2e^{-n^{1/6}/12}$. Thus, with probability $1 - e^{-\Omega(n^{1/6})}$, $|T \cap G| > 0$ and $|S \cap G| < \sqrt{n}$. Thus, $f(S) \geq 1/2$. If $|S| > n - n^{2/3}$, then $f(S) \geq (n - 2n^{2/3})/(2n) = n(1 - o(1))/(2n(1 - o(1))) = (1 - o(1))/2$.

## B.2 Concentration bounds from Section 3

- For all $S \subseteq N$ such that $|S| \geq n^{1/4}$, with probability $1 - e^{-\Omega(n^{1/4})}$ over $\mathcal{F}^M$, we have $|M \cap S| \geq 1$. Let $X_i$, for $i \in S$, indicate if $i \in M$. With $\mu = |S|(n/2 - \sqrt{n})/n$ and $\delta = 1/2$,

$$\Pr\left[\left|M \cap S| - |S| \cdot \frac{n/2 - \sqrt{n}}{n}\right| \geq |S| \cdot \frac{n/2 - \sqrt{n}}{2n}\right] \leq 2e^{-n^{1/4} \cdot \frac{n/2 - \sqrt{n}}{12n}} = e^{-\Omega(n^{1/4})}.$$

  Thus, $|M \cap S| \geq 1$ with probability at least $1 - e^{-\Omega(n^{1/4})}$.

- Let $S \subseteq N$ and $f_i \in \mathcal{F}^M$ such that $m = 0$, $|S| \geq n^{3/4}$, and $S$ independent of $i$, then

$$(1 - o(1)) \cdot |S| \leq \frac{n/2 + \sqrt{n}}{\sqrt{n}} \cdot b, \frac{n/2 + \sqrt{n}}{n/2} \cdot g \leq (1 + o(1)) \cdot |S|.$$

  Let $X_j$, for $j \in S$, indicate if $j \in B_i$. With $\mu = |S|(\sqrt{n}/(n/2 + \sqrt{n}))$ and $\delta = n^{-1/16}$,

$$\Pr\left[\left|g - \frac{\sqrt{n}}{n/2 + \sqrt{n}} \cdot |S|\right| \geq n^{-1/16} \cdot \frac{\sqrt{n}}{n/2 + \sqrt{n}} \cdot |S|\right] \leq 2e^{-n^{-1/8} \cdot \frac{\sqrt{n}}{3(n/2 + \sqrt{n})} \cdot n^{7/8}}$$

$$= e^{-\Omega(n^{1/4})}$$

  Similarly,

$$\Pr\left[\left|g - \frac{n/2}{n/2 + \sqrt{n}} \cdot |S|\right| \geq n^{-1/16} \cdot \frac{n/2}{n/2 + \sqrt{n}} \cdot |S|\right] \leq 2e^{-n^{-1/8} \cdot \frac{n/2}{3(n/2 + \sqrt{n})} \cdot n^{7/8}}$$

$$= e^{-\Omega(n^{3/4})}$$

  and we obtain the desired result.

## C Missing analysis from Sections 3.1 and 3.2

**Lemma 10.** *For all $f \in \mathcal{F}$, the function $f$ is submodular.*

*Proof.* We show that the marginal contribution $f_S(a) := f(S \cup \{a\}) - f(S)$ of an element $a \in N$ is decreasing.

$$a \in B, f_S(a) = \begin{cases} \frac{1}{2\sqrt{n}} & \text{if } m = 0 \\ -\frac{2}{n} & \text{otherwise} \end{cases}$$

$$a \in G, f_S(a) = \begin{cases} \frac{1}{2\sqrt{n}} & \text{if } m = 0 \text{ and } g \leq n^{1/4} \\ -\frac{2}{n} & \text{otherwise} \end{cases}$$

$$a \in M \text{ and } m = 0, f_S(a) = \frac{1}{2} - \frac{b}{n} - \frac{b}{2\sqrt{n}} - \frac{g}{n} - \begin{cases} \frac{g}{2\sqrt{n}} & \text{if } g < n^{1/4} \\ \frac{n^{1/4}}{2\sqrt{n}} - \frac{g - n^{1/4}}{n} & \text{if } g \geq n^{1/4} \end{cases}$$

$$= \frac{1}{2} - \frac{b}{n} - \frac{b}{2\sqrt{n}} - \begin{cases} \frac{g}{n} + \frac{g}{2\sqrt{n}} & \text{if } g < n^{1/4} \\ \frac{g - n^{1/4}}{n} + \frac{n^{1/4}}{n} + \frac{n^{1/4}}{2\sqrt{n}} - \frac{g - n^{1/4}}{n} & \text{if } g \geq n^{1/4} \end{cases}$$

$$= \frac{1}{2} - \frac{b}{n} - \frac{b}{2\sqrt{n}} - \begin{cases} \frac{g}{n} + \frac{g}{2\sqrt{n}} & \text{if } g < n^{1/4} \\ \frac{n^{1/4}}{n} + \frac{n^{1/4}}{2\sqrt{n}} & \text{if } g \geq n^{1/4} \end{cases}$$

$$a \in M \text{ and } m > 0, f_S(a) = 0$$

For $a \in G$ or $a \in B$ it is immediate that these marginal contributions are decreasing. For $a \in M$, note that

$$\frac{1}{2} - \frac{b}{n} - \frac{b}{2\sqrt{n}} - \begin{cases} \frac{g}{n} + \frac{g}{2\sqrt{n}} & \text{if } g < n^{1/4} \\ \frac{n^{1/4}}{n} + \frac{n^{1/4}}{2\sqrt{n}} & \text{if } g \geq n^{1/4} \end{cases} \geq \frac{1}{2} - \frac{\sqrt{n}}{n} - \frac{\sqrt{n}}{2\sqrt{n}} - \frac{n^{1/4}}{n} + \frac{n^{1/4}}{2\sqrt{n}} \geq 0$$

so $f_S(a)$ is also decreasing. $\square$

**Lemma 3.** *Let $\mathcal{F}$ be a family of functions and $\mathcal{F}' = \{f_1, \ldots, f_p\} \subseteq \mathcal{F}$ be a subfamily of functions drawn from some distribution. Assume the following two conditions hold:*

1. ***Indistinguishability.** For all $S \subseteq N$, w.p. $1 - e^{-\Omega(n^{1/4})}$ over $\mathcal{F}'$: for every $f_i, f_j \in \mathcal{F}'$,*

$$f_i(S) = f_j(S);$$

2. ***$\alpha$-gap.** Let $S_i^\star$ be a minimizer of $f_i$, then w.p. $1$ over $\mathcal{F}'$: for all $S \subseteq N$,*

$$\mathbb{E}_{f_i \sim U(\mathcal{F}')} [f_i(S) - f_i(S_i^\star)] \geq \alpha;$$

*Then, $\mathcal{F}$ is not $\alpha$-minimizable from strictly less than $e^{\Omega(n^{1/4})}$ samples over any distribution $\mathcal{D}$.*

*Proof.* We first claim that for any distribution $\mathcal{D}$, there exists a family of functions $F \subseteq \mathcal{F}$ such that with probability $1 - e^{-\Omega(n^{1/4})}$ over $S \sim \mathcal{D}$, $f_i(S) = f_j(S)$ for all $f_i, f_j \in F$. Let $I(\mathcal{F}', S)$ be the event that $f_i(S) = f_j(S)$ for all $f_i, f_j \in \mathcal{F}'$. Then,

$$\sum_{F \subseteq \mathcal{F}} \Pr[\mathcal{F}' = F] \Pr_{S \sim \mathcal{D}}[I(F, S)] = \sum_{F \subseteq \mathcal{F}} \Pr[\mathcal{F}' = F] \sum_{S \subseteq N} \mathbb{1}_{I(F,S)} \Pr[S \sim \mathcal{D}]$$

$$= \sum_{S \subseteq N} \Pr[S \sim \mathcal{D}] \sum_{F \subseteq \mathcal{F}} \Pr[\mathcal{F}' = F] \mathbb{1}_{I(F,S)}$$

$$= \sum_{S \subseteq N} \Pr[S \sim \mathcal{D}] \Pr_{\mathcal{F}'}[I(\mathcal{F}', S)]$$

$$\geq \min_{S \subseteq N} \Pr_{\mathcal{F}'}[I(\mathcal{F}', S)].$$

Thus, there exists some $F = \{f_1, \ldots, f_p\}$ such that
$$\Pr_{S \sim \mathcal{D}}[I(F, S)] \geq \min_{S \subseteq N} \Pr_{\mathcal{F}'}[I(\mathcal{F}', S)].$$

Since $\min_{S \subseteq N} \Pr_{\mathcal{F}'}[I(\mathcal{F}', S)] = 1 - e^{-\Omega(n^{1/4})}$. Then, by a union bound over the samples, $f_i(S) = f_j(S)$ for all $f_i, f_j \in F$ and for all samples $S$ with probability $1 - e^{-\Omega(n^{1/4})}$, and we assume this is the case, as well as the gap condition, for the remaining of the proof.

It follows that the choices of the algorithm given samples from $f_i$, $i \in [p]$, are independent of $i$. Pick $i \in [p]$ uniformly at random and consider the (possibly randomized) vector $S$ returned by the algorithm. Since $S$ is independent of $i$ and by the $\alpha$-gap condition,
$$\mathbb{E}_{f_i \sim U(\mathcal{F}')}[f_i(S) - f_i(S_i^\star)] \geq \alpha.$$

Thus, there exists at least one $f_i \in F$ such that for $f_i$, the algorithm is at least an additive factor $\alpha$ away from $f_i(S_i^\star)$. $\qquad\square$

# D   Missing analysis from Section 3.3

We begin by reviewing known results for classification and linear regression using the VC-dimension and the Rademacher complexity (Section D.1) and then give the missing analysis for the learning algorithm (Section D.2).

## D.1   VC-dimension and Rademacher complexities essentials

We review learning results needed for the analysis. These results use the VC-dimension and the Rademacher complexity, two of the most common tools to bound the generalization error of a learning algorithm. We formally define the VC-dimension and the Rademacher complexity using definitions from [SSBD14]. We begin with the VC-dimension, which is for classes of binary functions. We first define the concepts of restriction to a set and of shattering, which are useful to define the VC-dimension.

**Definition 11.** *(Restriction of $\mathcal{H}$ to $A$). Let $\mathcal{H}$ be a class of functions from $\mathcal{X}$ to $\{0,1\}$ and let $A = \{a_1, \ldots, a_m\} \subset \mathcal{X}$. The restriction of $\mathcal{H}$ to $A$ is the set of functions from $A$ to $\{0,1\}$ that can be derived from $\mathcal{H}$. That is,*
$$\mathcal{H}_A = \{(h(a_1), \ldots, h(a_m)) \ : \ h \in \mathcal{H}\},$$
*where we represent each function from $A$ to $\{0,1\}$ as a vector in $\{0,1\}^{|A|}$.*

**Definition 12.** *(Shattering). A hypothesis class $\mathcal{H}$ shatters a finite set $A \subset \mathcal{X}$ if the restriction of $\mathcal{H}$ to $A$ is the set of all functions from $A$ to $\{0,1\}$. That is, $|\mathcal{H}_A| = 2^{|A|}$.*

**Definition 13.** *(VC-dimension). The VC-dimension of a hypothesis class $\mathcal{H}$ is the maximal size of a set $S \subset \mathcal{X}$ that can be shattered by $\mathcal{H}$. If $\mathcal{H}$ can shatter sets of arbitrarily large size we say that $\mathcal{H}$ has infinite VC-dimension.*

Next, we define the Rademacher complexity, which is for more complex classes of functions than binary functions, such as real-valued functions.

**Definition 14.** *(Rademacher complexity). Let $\sigma$ be distributed i.i.d. according to $\Pr[\sigma_i = 1] = \Pr[\sigma_i = -1] = 1/2$. The Rademacher complexity $R(A)$ of set of vectors $A \subset \mathbb{R}^m$ is defined as*
$$R(A) := \frac{1}{m} \mathbb{E}_{\sigma}\left[\sup_{a \in A} \sum_{i=1}^{m} \sigma_i a_i\right].$$

This first result bounds the generalization error of a class of binary functions in terms of the VC-dimension of these classifiers.

**Theorem 15** ([SSBD14]). *Let $\mathcal{H}$ be a hypothesis class of functions from a domain $\mathcal{X}$ to $\{-1,1\}$ and $f : \mathcal{X} \mapsto \{-1,1\}$ be some "correct" function. Assume that the VC-dimension of $\mathcal{H}$ is $d$. Then, there is an absolute constant $C$ such that with $m \geq C(d + \log(1/\delta))/\epsilon^2$ i.i.d. samples $\mathbf{x}^1, \ldots, \mathbf{x}^m \sim \mathcal{D}$,*
$$\left|\Pr_{S \sim \mathcal{D}}[h(S) \neq f(S)] - \frac{1}{m}\sum_{i=1}^{m} \mathbb{1}_{h(S_i) \neq f(S_i)}\right| \leq \epsilon$$
*for all $h \in \mathcal{H}$, with probability $1 - \delta$ over the samples.*

We use the class of halfspaces for classification, for which we know the VC-dimension.

**Theorem 16** ([SSBD14])**.** *Let $b \in \mathbb{R}$. The class of functions $\{\mathbf{x} \mapsto \text{sign}(\mathbf{w}^\intercal \mathbf{x}) + b : \mathbf{w} \in \mathbb{R}^n\}$ has VC dimension $n$.*

The following result combines the generalization error of a class of functions in terms of its Rademacher complexity with the Rademacher complexity of linear functions over a $\rho$-Lipschitz loss function.

**Theorem 17** ([SSBD14])**.** *Suppose that $\mathcal{D}$ is a distribution over $\mathcal{X}$ such that with probability 1 over $\mathbf{x} \sim \mathcal{D}$ we have that $\|\mathbf{x}\|_\infty \leq R$. Let $\mathcal{H} = \{\mathbf{w} \in \mathbb{R}^d : \|\mathbf{w}\|_1 \leq B\}$ and let $\ell(\mathbf{w}, (\mathbf{x}, y)) = \phi(\mathbf{w}^\intercal \mathbf{x}, y)$ such that for all $y \in \mathcal{Y}$, $a \mapsto \phi(a, y)$ is an $\rho$-Lipschitz function and such that $\max_{a \in [-BR, BR]} |\phi(a, y)| \leq c$. Then, for any $\delta \in (0, 1)$, with probability of at least $1 - \delta$ over the choice of an i.i.d. sample of size $m$,*

$$\mathbb{E}_{\mathbf{x} \sim \mathcal{D}}[|\ell(\mathbf{w}, (\mathbf{x}, y))|] \leq \frac{1}{|\mathcal{S}|} \sum_{\mathbf{x} \in \mathcal{S}} |\ell(\mathbf{w}, (\mathbf{x}, y))| + 2\rho B R \sqrt{\frac{2 \log(2d)}{m}} + c \sqrt{\frac{2 \log(2/\delta)}{m}}$$

*for all $\mathbf{w} \in \mathcal{H}$.*

### D.2 Missing analysis for the learning algorithm

We formally define `PAC` learnability with absolute loss.

**Definition 18** (`PAC` learnability with absolute loss)**.** *A class of functions $\mathcal{F}$ is `PAC` learnable if there exist a function $m_\mathcal{F} : (0, 1)^2 \to \mathbb{N}$ and a learning algorithm with the following property: For every $\epsilon, \delta \in (0, 1)$, for every distribution $\mathcal{D}$, and every function $f \in \mathcal{F}$, when running the learning algorithm on $m \geq m_\mathcal{H}(\epsilon, \delta)$ i.i.d. samples $(S, f(S))$ with $S \sim \mathcal{D}$, the algorithm returns a function $\tilde{f}$ such that, with probability at least $1 - \delta$ (over the choice of the $m$ training examples),*

$$\mathbb{E}_{S \sim \mathcal{D}}\left[\left|\tilde{f}(S) - f(S)\right|\right] \leq \epsilon.$$

For the remaining of this section, let $f \in \mathcal{F}$ be the function defined over partition $(G, B, M)$ for which the learning algorithm is given samples $\mathcal{S}$. Let $\mathbf{x}_S$ denote the $0 - 1$ vector corresponding to the set $S$, i.e., $x_i = \mathbb{1}_{i \in S}$. We define the following subcollection of samples, $\mathcal{S}_{\tilde{\mathcal{Z}}_i} := \{(S, f(S)) \in \mathcal{S} : S \in \tilde{\mathcal{Z}}_i\}$, $\mathcal{S}_{\tilde{\mathcal{X}} \cup \tilde{\mathcal{Y}}}^{\leq} := \{(S_j, f(S_j)) \in \mathcal{S} : S_j \notin \tilde{\mathcal{Z}}, j \leq m/2\}$, $\mathcal{S}_{\tilde{\mathcal{Y}}}^{>} := \{(S_j, f(S_j)) \in \mathcal{S} : S_j \in \tilde{\mathcal{Y}}, j > m/2\}$. Let $\mathcal{R}$ be a region of sets, we define $\mathcal{D}_\mathcal{R}$ to be the distribution $S \sim \mathcal{D}$ conditioned on $S \in \mathcal{R}$. The linear regression predictors $f_{\tilde{\mathcal{Z}}_i}$ and $f_{\tilde{\mathcal{Y}}}$ and the classifier $C$ are formally defined as follows.

$$\tilde{\mathbf{w}}_i := \underset{\mathbf{w} \in \mathbb{R}^n : \|\mathbf{w}\|_1 \leq 1}{\text{argmin}} \sum_{(S_j, f(S_j)) \in \mathcal{S}_{\tilde{\mathcal{Z}}_i}} |\mathbf{w}^\intercal \mathbf{x}_S + 1 - f(S)|$$

$$\tilde{\mathbf{w}}_C := \underset{\mathbf{w} \in \mathbb{R}^n}{\text{argmin}} \sum_{(S_j, f(S_j)) \in \mathcal{S}_{\tilde{\mathcal{X}} \cup \tilde{\mathcal{Y}}}^{\leq}} \mathbb{1}_{\text{sign}\left(\mathbf{w}^\intercal \mathbf{x}_S + n^{1/4}\right) = \text{sign}(f(S) - |S|/(2\sqrt{n}))}$$

$$\tilde{\mathbf{w}}_{\tilde{\mathcal{Y}}} := \underset{\mathbf{w} \in \mathbb{R}^n : \|\mathbf{w}\|_1 \leq 1}{\text{argmin}} \sum_{(S_j, f(S_j)) \in \mathcal{S}_{\tilde{\mathcal{Y}}}^{>}} |\mathbf{w}^\intercal \mathbf{x}_S + b_\mathcal{Y} - f(S)|$$

where $b_\mathcal{Y}$ is the constant $b_\mathcal{Y} = 1/2 + 1/(2n^{1/4}) + 1/n^{3/4}$. Then,

$$f_{\tilde{\mathcal{Z}}_i}(S) := \tilde{\mathbf{w}}_i^\intercal \cdot \mathbf{x}_S + 1$$

$$C(S) := \text{sign}\left(\tilde{\mathbf{w}}_C^\intercal \mathbf{x}_S + n^{1/4}\right)$$

$$f_{\tilde{\mathcal{Y}}}(S) := \tilde{\mathbf{w}}_{\tilde{\mathcal{Y}}}^\intercal \cdot \mathbf{x}_S + b_\mathcal{Y}$$

We show that there are no false negatives for $\tilde{Z}$, which is important for the existence of a linear classifier $C$ with zero empirical error on samples $\mathcal{S}_{\tilde{X} \cup \tilde{Y}}^{\leq}$.

**Lemma 19.** *If $S \notin \tilde{\mathcal{Z}}$, then $S \in \mathcal{X} \cup \mathcal{Y}$.*

*Proof.* The proof is by contrapositive. If $S \notin \mathcal{X} \cup \mathcal{Y}$, then there exists $i \in M$ such that $i \in S$. Then note that $f(S)$ is an affine function over all $S$ such that $i \in S$. So it must be the case that the empirical minimizer $\tilde{\mathbf{w}}_i$ computed has zero empirical loss. Thus $i \in \tilde{M}$. $\qquad\square$

We give a general lemma for the expected error of a linear regression predictor.

**Lemma 20.** *Assume $\mathcal{S}'$ is a collection of $m \geq \epsilon^{-2}(\log(2n) + \log(2/\delta))/2$ i.i.d. samples from some distribution $\mathcal{D}'$. Then, with probability at least $1 - \delta$,*

$$\mathop{\mathbb{E}}_{S \sim \mathcal{D}'} \left[|\mathbf{w}^\mathsf{T}\mathbf{x}_S + b - f(S)|\right] \leq \frac{1}{|\mathcal{S}'|} \sum_{S \in \mathcal{S}'} |\mathbf{w}^\mathsf{T}\mathbf{x}_S + b - f(S)| + \epsilon$$

*for all $\mathbf{w} \in \mathbb{R}^n$ such that $\|\mathbf{w}\|_1 \leq 1$.*

*Proof.* Consider the setting of Theorem 17 with the absolute loss, so $\ell(\mathbf{w}, (\mathbf{x}_S, f(S)) = \phi(\mathbf{w}^\mathsf{T}\mathbf{x}_S, f(S)) = |\mathbf{w}^\mathsf{T}\mathbf{x}_S + b - f(S)|$ for some $b \in \mathbb{R}$. Notice that the absolute loss is 1-Lipschitz, so $\rho = 1$. We also have $d = n$ and $\|\mathbf{x}_S\|_\infty \leq 1 = R$ for all $S$. We consider $\|\mathbf{w}\|_1 \leq 1 = B$. Such $B$ and $R$ imply that $c = 2$ satisfies the condition of Theorem 17. Thus, if $\mathcal{S}'$ is a set of $m \geq \epsilon^{-2}(\log(2n) + \log(2/\delta))/2$ i.i.d. samples from a distribution $\mathcal{D}'$, then with probability at least $1 - \delta$,

$$\mathop{\mathbb{E}}_{S \sim \mathcal{D}'} \left[|\tilde{\mathbf{w}}^\mathsf{T}\mathbf{x}_S + b - f(S)|\right] \leq \frac{1}{|\mathcal{S}'|} \sum_{S \in \mathcal{S}'} \left|\tilde{\mathbf{w}}_{\mathcal{Y}}^\mathsf{T}\mathbf{x}_S + b - f(S)\right| + \epsilon$$

for all $\mathbf{w} \in \mathbb{R}^n$ such that $\|\mathbf{w}\|_1 \leq 1$. $\qquad\square$

We show that if $S \in \tilde{Z}_i$ with $i \in \tilde{M}$, then the learner $\tilde{f}$ directs $S$ to $f_{\tilde{\mathcal{Z}}_i}$.

**Lemma 21.** *Assume $S \in \tilde{\mathcal{Z}}_i$, with $i \in \tilde{M}$. Then, $\tilde{f}(S) = f_{\tilde{\mathcal{Z}}_i}(S)$*

*Proof.* We argue that $i = \min(\{i' : i \in S \cap \tilde{M}\})$. Assume by contradiction that there exists $i' < i$ such that $i' \in S \cap \tilde{M}$. Then it must be the case that $S \in \tilde{\mathcal{Z}}$ before iteration $i$ of the algorithm. But then, $S$ would not have been considered for $\tilde{\mathcal{Z}}_i$. $\qquad\square$

The following lemma bounds the error of linear regression predictors $f_{\tilde{\mathcal{Z}}_i}$ with $i \in \tilde{M}$.

**Lemma 22.** *Assume $|\mathcal{S}_{\tilde{\mathcal{Z}}_i}| \geq \epsilon^{-2}(\log(2n) + \log(2/\delta))/2$ and $i \in \tilde{M}$. Then with probability $1 - \delta$ over $\mathcal{S}_{\tilde{\mathcal{Z}}_i}$,*

$$\mathop{\mathbb{E}}_{S \sim \mathcal{D}_{\tilde{\mathcal{Z}}_i}} \left[\left|\tilde{f}(S) - f(S)\right|\right] = \mathop{\mathbb{E}}_{S \sim \mathcal{D}_{\tilde{\mathcal{Z}}_i}} \left[|\tilde{\mathbf{w}}_i^\mathsf{T}\mathbf{x}_S + 1 - f(S)|\right] \leq \epsilon.$$

*Proof.* By Lemma 21,

$$\mathop{\mathbb{E}}_{S \sim \mathcal{D}_{\tilde{\mathcal{Z}}_i}} \left[\left|\tilde{f}(S) - f(S)\right|\right] = \mathop{\mathbb{E}}_{S \sim \mathcal{D}_{\tilde{\mathcal{Z}}_i}} \left[|\tilde{\mathbf{w}}_i^\mathsf{T}\mathbf{x}_S + 1 - f(S)|\right]$$

Let $i \in \tilde{M}$, then, the empirical loss of $\tilde{\mathbf{w}}_i$ is zero, i.e., $\sum_{S \in \mathcal{S}_i} |\tilde{\mathbf{w}}_i^\mathsf{T}\mathbf{x}_S + 1 - f(S)| = 0$ by Algorithm 1 since $i \in \tilde{M}$. The collection of samples $\mathcal{S}_{\tilde{\mathcal{Z}}_i}$ consists of $m$ i.i.d. samples $S$ from $\mathcal{D}_{\tilde{\mathcal{Z}}_i}$, so by Lemma 20,

$$\mathop{\mathbb{E}}_{S \sim \mathcal{D}_{\tilde{\mathcal{Z}}_i}} \left[|\tilde{\mathbf{w}}_i^\mathsf{T}\mathbf{x}_S + 1 - f(S)|\right] \leq \epsilon.$$

$\qquad\square$

Next, we bound the error of classifier $C$.

**Lemma 23.** *Assume $|\mathcal{S}_{\tilde{\mathcal{X}} \cup \tilde{\mathcal{Y}}}^{\lessgtr}| \geq C(n + \log(1/\delta))/\epsilon^2$. Then, with probability $1 - \delta$ over $\mathcal{S}_{\tilde{\mathcal{X}} \cup \tilde{\mathcal{Y}}}^{\lessgtr}$,*

$$\mathop{\Pr}_{S \sim \mathcal{D}_{\tilde{\mathcal{X}} \cup \tilde{\mathcal{Y}}}} \left[\text{sign}\left((\tilde{\mathbf{w}}_C)^\mathsf{T}\mathbf{x}_S + n^{1/4}\right) = \text{sign}\left(S \in \mathcal{X}\right)\right] \leq \epsilon.$$

*Proof.* Note that the support of $\mathcal{D}_{\tilde{\mathcal{X}} \cup \tilde{\mathcal{Y}}}$ does not contain any set $S \in \mathcal{Z}$ by Lemma 19. Thus we only consider $S \in \mathcal{X} \cup \mathcal{Y}$ for the remaining of the analysis. Consider the classifier

$$h^{\star}(S) = \text{sign}\left(({\mathbf{w}^{\star}})^{\mathsf{T}} \mathbf{x}_S + n^{1/4}\right)$$

where $w_i^{\star} = -1$ if $i \in G$ and $w_i^{\star} = 0$ otherwise. Then $h^{\star}(S) = 1$ if $S \in \mathcal{X}$ and $h^{\star}(S) = -1$ if $S \in \mathcal{Y}$. Since $\mathbf{w}^{\star}$ has zero empirical error over $\mathcal{S}_{\tilde{\mathcal{X}} \cup \tilde{\mathcal{Y}}}^{\leq}$, it must also be the case for the empirical risk minimizer $\tilde{\mathbf{w}}_C$,

$$\sum_{(S, f(S)) \in \mathcal{S}_{\tilde{\mathcal{X}} \cup \tilde{\mathcal{Y}}}^{\leq}} \mathbb{1}_{\text{sign}\left((\tilde{\mathbf{w}}_C)^{\mathsf{T}} \mathbf{x}_S + n^{1/4}\right) = \text{sign}(S \in \mathcal{X})} = 0.$$

By Theorem 16, the class of functions $\{S \mapsto \text{sign}(\mathbf{w}^{\mathsf{T}} \mathbf{x} + n^{1/4}) : \mathbf{w} \in \mathbb{R}^n\}$ has VC dimension $n$. Since $\mathcal{S}_{\tilde{\mathcal{X}} \cup \tilde{\mathcal{Y}}}^{\leq}$ is a collection of i.i.d. samples $S$ from $\mathcal{D}_{\tilde{\mathcal{X}} \cup \tilde{\mathcal{Y}}}$, we conclude by Theorem 15 that with probability $1 - \delta$ over $\mathcal{S}_{\tilde{\mathcal{X}} \cup \tilde{\mathcal{Y}}}^{\leq}$,

$$\Pr_{S \sim \mathcal{D}_{\tilde{\mathcal{X}} \cup \tilde{\mathcal{Y}}}}\left[\text{sign}\left((\tilde{\mathbf{w}}_C)^{\mathsf{T}} \mathbf{x}_S + n^{1/4}\right) = \text{sign}\left(S \in \mathcal{X}\right)\right] \leq \epsilon$$

with $|\mathcal{S}_{\tilde{\mathcal{X}} \cup \tilde{\mathcal{Y}}}^{\leq}| \geq C(n + \log(1/\delta))/\epsilon^2$.

$\square$

We bound the error of the linear regression predictor $\tilde{\mathbf{w}}_{\tilde{Y}}$. Note the additional $\frac{|\mathcal{S}_{\tilde{\mathcal{Y}}}^{\geq} \cap \mathcal{X}|}{|\mathcal{S}_{\tilde{\mathcal{Y}}}^{\geq}|}$ term that is due the predictor being trained over $\mathcal{S}_{\tilde{\mathcal{Y}}}^{\geq}$ which might contain samples in $\mathcal{X}$ which have been misclassified in $\tilde{\mathcal{Y}}$.

**Lemma 24.** *Assume* $|\mathcal{S}_{\tilde{\mathcal{Y}}}^{\geq}| \geq \epsilon^{-2}(\log(2n) + \log(2/\delta))/2$. *Then with probability* $1 - \delta$ *over* $\mathcal{S}_{\tilde{\mathcal{Y}}}^{\geq}$,

$$\mathbb{E}_{S \sim \mathcal{D}_{\tilde{\mathcal{Y}}}}\left[\left|\tilde{\mathbf{w}}_{\tilde{\mathcal{Y}}}^{\mathsf{T}} \mathbf{x}_S + b_{\mathcal{Y}} - f(S)\right|\right] \leq \epsilon + \frac{|\mathcal{S}_{\tilde{\mathcal{Y}}}^{\geq} \cap \mathcal{X}|}{|\mathcal{S}_{\tilde{\mathcal{Y}}}^{\geq}|}.$$

*Proof.* Consider the affine function $(\mathbf{w}_{\mathcal{Y}}, b)$ over all sets in region $\mathcal{Y}$. This function has empirical loss:

$$\frac{1}{|\mathcal{S}_{\tilde{\mathcal{Y}}}^{\geq}|} \sum_{(S, f(S)) \in \mathcal{S}_{\tilde{\mathcal{Y}}}^{\geq}} \left|\mathbf{w}_{\mathcal{Y}}^{\mathsf{T}} \mathbf{x}_S + b_{\mathcal{Y}} - f(S)\right|$$

$$= \frac{1}{|\mathcal{S}_{\tilde{\mathcal{Y}}}^{\geq}|} \left( \sum_{(S, f(S)) \in \mathcal{S}_{\tilde{\mathcal{Y}}}^{\geq} : S \in \mathcal{X}} \left|\mathbf{w}_{\mathcal{Y}}^{\mathsf{T}} \mathbf{x}_S + b_{\mathcal{Y}} - f(S)\right| + \sum_{(S, f(S)) \in \mathcal{S}_{\tilde{\mathcal{Y}}}^{\geq} : S \in \mathcal{Y}} \left|\mathbf{w}_{\mathcal{Y}}^{\mathsf{T}} \mathbf{x}_S + b_{\mathcal{Y}} - f(S)\right| \right)$$

$$\leq \frac{1}{|\mathcal{S}_{\tilde{\mathcal{Y}}}^{\geq}|} \left( |\mathcal{S}_{\tilde{\mathcal{Y}}}^{\geq} \cap \mathcal{X}| + 0 \right)$$

$$= \frac{|\mathcal{S}_{\tilde{\mathcal{Y}}}^{\geq} \cap \mathcal{X}|}{|\mathcal{S}_{\tilde{\mathcal{Y}}}^{\geq}|}$$

Since $\tilde{\mathbf{w}}_{\tilde{\mathcal{Y}}}$ is the empirical loss minimizer, it has smaller empirical loss than $\mathbf{w}_{\mathcal{Y}}$, so $\sum_{(S, f(S)) \in \mathcal{S}_{\tilde{\mathcal{Y}}}^{\geq}} \left|\tilde{\mathbf{w}}_{\tilde{\mathcal{Y}}}^{\mathsf{T}} \mathbf{x}_S + b_{\mathcal{Y}} - f(S)\right| \leq \frac{|\mathcal{S}_{\tilde{\mathcal{Y}}}^{\geq} \cap \mathcal{X}|}{|\mathcal{S}_{\tilde{\mathcal{Y}}}^{\geq}|}$.

Since $\mathcal{S}_{\tilde{\mathcal{Y}}}^{\geq}$ consists of $m$ i.i.d. samples from $\mathcal{D}_{\tilde{\mathcal{Y}}}$ (in particular, none of these samples were used to train classifier $C$) and by Theorem 20, we conclude that

$$\mathbb{E}_{S \sim \mathcal{D}_{\tilde{\mathcal{Y}}}}\left[\left|\tilde{\mathbf{w}}_{\tilde{\mathcal{Y}}}^{\mathsf{T}} \mathbf{x}_S + b_{\mathcal{Y}} - f(S)\right|\right] \leq \epsilon + \frac{|\mathcal{S}_{\tilde{\mathcal{Y}}}^{\geq} \cap \mathcal{X}|}{|\mathcal{S}_{\tilde{\mathcal{Y}}}^{\geq}|}.$$

$\square$

We show a general lemma to analyze the cases where the linear predictors are trained over a small number of samples.

**Lemma 25.** *Let $\mathcal{S}$ be the collection of $m$ i.i.d. samples drawn from $\mathcal{D}$. Assume $m \geq \frac{12}{\epsilon} \log(2/\delta)$ and let $\mathcal{S}' \subseteq \mathcal{S}$, then it is either the case that*

$$\Pr_{S \sim \mathcal{D}} [S \in \mathcal{S}'] \leq \epsilon$$

*or*

$$|\mathcal{S} \cap \mathcal{S}'| \geq \frac{\epsilon m}{2}$$

*with probability at least $1 - \delta$.*

*Proof.* By Chernoff bound with $\mu = \epsilon m$ and $m \geq \frac{12}{\epsilon} \log(2/\delta)$:

$$\Pr_{S} [|\mathcal{S} \cap \mathcal{S}'| - \epsilon m| \geq \epsilon m/2] \leq 2e^{-\epsilon m/12} \leq \delta$$

$\square$

We bound the additional $\frac{|\mathcal{S}_{\tilde{y}}^{\geq} \cap \mathcal{X}|}{|\mathcal{S}_{\tilde{y}}^{\geq}|}$ term from Lemma 24 in the case where the number of samples $\mathcal{S}_{\tilde{y}}^{\geq}$ is large enough.

**Lemma 26.** *Assume $|\mathcal{S}_{\tilde{y}}^{\geq}| \geq 12n^2 \log(2n/\delta)/\epsilon^2$, then*

$$\frac{|\mathcal{S}_{\tilde{y}}^{\geq} \cap \mathcal{X}|}{|\mathcal{S}_{\tilde{y}}^{\geq}|} \leq \frac{3}{2} \Pr_{S \sim \mathcal{D}_{\tilde{y}}} [S \in \mathcal{X}] + \frac{3\epsilon}{2n}$$

*with probability $1 - \delta/n$*

*Proof.* If $\Pr_{S \sim \mathcal{D}_{\tilde{y}}} [S \in \mathcal{X}] \geq \epsilon/n$, then by the Chernoff bound:

$$\Pr \left[ ||\mathcal{S}_{\tilde{y}}^{\geq} \cap \mathcal{X}| - |\mathcal{S}_{\tilde{y}}^{\geq}| \Pr_{S \sim \mathcal{D}_{\tilde{y}}} [S \in \mathcal{X}]| \geq \frac{1}{2} \Pr_{S \sim \mathcal{D}_{\tilde{y}}} [S \in \mathcal{X}] |\mathcal{S}_{\tilde{y}}^{\geq}| \right] \leq 2e^{-|\mathcal{S}_{\tilde{y}}^{\geq}| \Pr_{S \sim \mathcal{D}_{\tilde{y}}}[S \in \mathcal{X}]/12}$$

$$\leq 2e^{-|\mathcal{S}_{\tilde{y}}^{\geq}| \epsilon/(12n)}$$

and $|\mathcal{S}_{\tilde{y}}^{\geq} \cap \mathcal{X}| \leq \frac{3}{2} |\mathcal{S}_{\tilde{y}}^{\geq}| \Pr_{S \sim \mathcal{D}_{\tilde{y}}} [S \in \mathcal{X}]$ with probability $1 - \delta/n$ since $|\mathcal{S}_{\tilde{y}}^{\geq}| \geq 12n^2 \log(2n/\delta)/\epsilon^2$. Otherwise, by another Chernoff bound:

$$\Pr \left[ ||\mathcal{S}_{\tilde{y}}^{\geq} \cap \mathcal{X}| - |\mathcal{S}_{\tilde{y}}^{\geq}| \Pr_{S \sim \mathcal{D}_{\tilde{y}}} [S \in \mathcal{X}]| \geq \frac{\epsilon}{2n} \cdot |\mathcal{S}_{\tilde{y}}^{\geq}| \right]$$

$$= \Pr \left[ ||\mathcal{S}_{\tilde{y}}^{\geq} \cap \mathcal{X}| - |\mathcal{S}_{\tilde{y}}^{\geq}| \Pr_{S \sim \mathcal{D}_{\tilde{y}}} [S \in \mathcal{X}]| \geq \frac{1}{2n} \frac{\epsilon}{\Pr_{S \sim \mathcal{D}_{\tilde{y}}} [S \in \mathcal{X}]} \cdot |\mathcal{S}_{\tilde{y}}^{\geq}| \Pr_{S \sim \mathcal{D}_{\tilde{y}}} [S \in \mathcal{X}] \right]$$

$$\leq 2e^{-|\mathcal{S}_{\tilde{y}}^{\geq}| \Pr_{S \sim \mathcal{D}_{\tilde{y}}}[S \in \mathcal{X}](\frac{1}{2n} \frac{\epsilon}{\Pr_{S \sim \mathcal{D}_{\tilde{y}}}[S \in \mathcal{X}]})^2/3}$$

$$\leq 2e^{-|\mathcal{S}_{\tilde{y}}^{\geq}| \epsilon^2/(12n^2)}$$

and since $\Pr_{S \sim \mathcal{D}_{\tilde{y}}} [S \in \mathcal{X}] \leq \epsilon/n$, $|\mathcal{S}_{\tilde{y}}^{\geq} \cap \mathcal{X}| \leq 3|\mathcal{S}_{\tilde{y}}^{\geq}| \epsilon/(2n)$ with probability $1 - \delta/n$ since $|\mathcal{S}_{\tilde{y}}^{\geq}| \geq 12n^2 \log(2n/\delta)/\epsilon^2$ $\square$

We are now ready to put all the pieces together and show the main result for the learning algorithm.

**Lemma 7.** *Let $\tilde{f}$ be the predictor returned by Algorithm 1, then w.p. $1 - \delta$ over $m \in O(n^3 + n^2(\log(2n/\delta))/\epsilon^2)$ samples $\mathcal{S}$ drawn i.i.d. from any distribution $\mathcal{D}$, $\mathbb{E}_{S \sim \mathcal{D}}[|\tilde{f}(S) - f(S)|] \leq \epsilon$.*

*Proof.* Let $\ell(S) = |\tilde{f}(S) - f(S)|$ be the loss by the algorithm. We divide the analysis in three cases dependent on which region $S \sim \mathcal{D}$ falls into.

$$\mathbb{E}_{S \sim \mathcal{D}} [\ell(S)] = \Pr_{S \sim \mathcal{D}} [S \in \tilde{\mathcal{X}}] \mathbb{E}_{S \sim \mathcal{D}_{\tilde{\mathcal{X}}}} [\ell(S)] + \Pr_{S \sim \mathcal{D}} [S \in \tilde{\mathcal{Y}}] \mathbb{E}_{S \sim \mathcal{D}_{\tilde{y}}} [\ell(S)] + \sum_{i \in \tilde{M}} \Pr_{S \sim \mathcal{D}} [S \in \tilde{\mathcal{Z}}_i] \mathbb{E}_{S \sim \mathcal{D}_{\tilde{\mathcal{Z}}_i}} [\ell(S)]$$

We analyze each of these three cases separately. Note that since $f(S) \in [0,1]$ and since all the linear regression predictors $\mathbf{w}$ have bounded norm $\|\mathbf{w}\|_1 \le 1$, $\ell(S) \le 2$.

If $S \in \tilde{\mathcal{X}}$. Then, $S \notin \mathcal{Z}$ by Lemma 19. Thus it is either the case that $S \in \mathcal{X}$ or $S \in \mathcal{Y}$. We get

$$\mathbb{E}_{S \sim \mathcal{D}_{\tilde{\mathcal{X}}}}[\ell(S)] = \Pr_{S \sim \mathcal{D}_{\tilde{\mathcal{X}}}}[S \in \mathcal{X}] \mathbb{E}_{S \sim \mathcal{D}_{\mathcal{X} \cap \tilde{\mathcal{X}}}}[\ell(S)] + \Pr_{S \sim \mathcal{D}_{\tilde{\mathcal{X}}}}[S \in \mathcal{Y}] \mathbb{E}_{S \sim \mathcal{D}_{\tilde{\mathcal{X}} \cap \mathcal{Y}}}[\ell(S)]$$

$$\le 1 \cdot \mathbb{E}_{S \sim \mathcal{D}_{\mathcal{X} \cap \tilde{\mathcal{X}}}}\left[\frac{|S|}{2\sqrt{n}} - \frac{|S|}{2\sqrt{n}}\right] + \Pr_{S \sim \mathcal{D}_{\tilde{\mathcal{X}}}}[S \in \mathcal{Y}] \cdot 2$$

$$\le 2 \Pr_{S \sim \mathcal{D}_{\tilde{\mathcal{X}}}}[S \in \mathcal{Y}]$$

If $S \in \tilde{\mathcal{Y}}$. Then, if $\Pr_{S \sim \mathcal{D}}[S \in \tilde{\mathcal{Y}}] \le \epsilon/(2n)$,

$$\Pr_{S \sim \mathcal{D}}[S \in \tilde{\mathcal{Y}}] \mathbb{E}_{S \sim \mathcal{D}_{\tilde{\mathcal{Y}}}}[\ell(S)] \le \frac{\epsilon}{n}$$

Otherwise, by Lemma 25, $|\mathcal{S}_{\tilde{\mathcal{Y}}}^{>}| \ge \epsilon m/(4n) \ge (n/\epsilon)^2 \log(2n/\delta)$, so by Lemma 24, with probability at least $1 - \delta/n$,

$$\Pr_{S \sim \mathcal{D}}[S \in \tilde{\mathcal{Y}}] \mathbb{E}_{S \sim \mathcal{D}_{\tilde{\mathcal{Y}}}}[\ell(S)] \le \frac{\epsilon}{n} + \Pr_{S \sim \mathcal{D}}[S \in \tilde{\mathcal{Y}}]\left(\frac{|\mathcal{S}_{\tilde{\mathcal{Y}}}^{>} \cap \mathcal{X}|}{|\mathcal{S}_{\tilde{\mathcal{Y}}}^{>}|}\right)$$

If $S \in \tilde{\mathcal{Z}}_i$. Then if $\Pr_{S \sim \mathcal{D}}[S \in \mathcal{Z}_i] \le \epsilon/(4n)$,

$$\Pr_{S \sim \mathcal{D}}[S \in \tilde{\mathcal{Z}}_i] \mathbb{E}_{S \sim \mathcal{D}_i}[\ell(S)] \le \frac{\epsilon}{2n}.$$

Otherwise, by Lemma 25 $|\mathcal{S}_i| \ge \epsilon m/(8n) \ge (2n/\epsilon)^2 \log(2n/\delta)$, so by Lemma 22, with probability at least $1 - \delta/n$,

$$\Pr_{S \sim \mathcal{D}}[S \in \tilde{\mathcal{Z}}_i] \mathbb{E}_{S \sim \mathcal{D}_i}[\ell(S)] \le \frac{\epsilon}{2n}.$$

Combining the three cases, and by a union bound over the at most $n$ events with probability at least $1 - \delta/n$, we have that with probaiblity at least $1 - \delta$,

$$\mathbb{E}_{S \sim \mathcal{D}}[\ell(S)]$$

$$= \Pr_{S \sim \mathcal{D}}[S \in \tilde{\mathcal{X}}] \mathbb{E}_{S \sim \mathcal{D}_{\tilde{\mathcal{X}}}}[\ell(S)] + \Pr_{S \sim \mathcal{D}}[S \in \tilde{\mathcal{Y}}] \mathbb{E}_{S \sim \mathcal{D}_{\tilde{\mathcal{Y}}}}[\ell(S)] + \sum_{i \in \tilde{M}} \Pr_{S \sim \mathcal{D}}[S \in \tilde{\mathcal{Z}}_i] \mathbb{E}_{S \sim \mathcal{D}_{\tilde{\mathcal{Z}}_i}}[\ell(S)]$$

$$\le 2 \cdot \Pr_{S \sim \mathcal{D}}[S \in \tilde{\mathcal{X}}] \Pr_{S \sim \mathcal{D}_{\tilde{\mathcal{X}}}}[S \in \mathcal{Y}] + \Pr_{S \sim \mathcal{D}}[S \in \tilde{\mathcal{Y}}]\left[\frac{\epsilon}{n} + \frac{|\mathcal{S}_{\tilde{\mathcal{Y}}}^{>} \cap \mathcal{X}|}{|\mathcal{S}_{\tilde{\mathcal{Y}}}^{>}|}\right] + n \cdot \frac{\epsilon}{2n}$$

$$\le 2 \cdot \Pr_{S \sim \mathcal{D}}[S \in \tilde{\mathcal{X}}] \Pr_{S \sim \mathcal{D}_{\tilde{\mathcal{X}}}}[S \in \mathcal{Y}] + \Pr_{S \sim \mathcal{D}}[S \in \tilde{\mathcal{Y}}]\left[\frac{\epsilon}{n} + \frac{3}{2}\Pr_{S \sim \mathcal{D}_{\tilde{\mathcal{Y}}}}[S \in \mathcal{X}] + \frac{3\epsilon}{2n}\right] + \frac{\epsilon}{2}$$

$$\le 2 \cdot \Pr_{S \sim \mathcal{D}}[S \in \tilde{\mathcal{X}} \cup \tilde{\mathcal{Y}}] \Pr_{S \sim \mathcal{D}_{\tilde{\mathcal{X}} \cup \tilde{\mathcal{Y}}}}[S \in (\tilde{\mathcal{X}} \cap \mathcal{Y}) \cup (\tilde{\mathcal{Y}} \cap \mathcal{X})] + \frac{3\epsilon}{4}$$

where the third inequality is by Lemma 26. If $\Pr_{S \sim D}[S \in \tilde{\mathcal{X}} \cup \tilde{\mathcal{Y}}] \le \frac{\epsilon}{n}$,

$$\Pr_{S \sim \mathcal{D}}[S \in \tilde{\mathcal{X}} \cup \tilde{\mathcal{Y}}] \Pr_{S \sim \mathcal{D}_{\tilde{\mathcal{X}} \cup \tilde{\mathcal{Y}}}}[S \in (\tilde{\mathcal{X}} \cap \mathcal{Y}) \cup (\tilde{\mathcal{Y}} \cap \mathcal{X})] \le \frac{\epsilon}{n}.$$

Otherwise, by Lemma 25, $|\mathcal{S}_{\tilde{\mathcal{X}} \cup \tilde{\mathcal{Y}}}^{\le}| \ge \epsilon m/2n \ge c_1(n^3 + n^2 \log(n/\delta))/\epsilon^2$ with probability $1 - \delta/n$, so by Lemma 23,

$$\Pr_{S \sim \mathcal{D}_{\tilde{\mathcal{X}} \cup \tilde{\mathcal{Y}}}}[S \in (\tilde{\mathcal{X}} \cap \mathcal{Y}) \cup (\tilde{\mathcal{Y}} \cap \mathcal{X})] = \Pr_{S \sim \mathcal{D}_{\tilde{\mathcal{X}} \cup \tilde{\mathcal{Y}}}}[\text{sign}\left((\tilde{\mathbf{w}}_C)^\mathsf{T} \mathbf{x}_S + n^\epsilon\right) \ne \text{sign}\left(S \in \mathcal{X}\right)] \le \frac{\epsilon}{n}.$$

and we conlude that

$$\mathbb{E}_{S \sim \mathcal{D}}\left[\left|\tilde{f}(S) - f(S)\right|\right] \le \epsilon$$

with probability $1 - \delta$. $\qquad\square$