[Reviews · NeurIPS 2017]

Reviewer 1



In this paper, the authors discuss the problem of minimizing a submodular function `f' from samples (S_i,f(S_i)), where the hardness of this problem is described despite its polynomial-tractability in optimization. I think this paper is a natural and necessary expansion of this type of problem from submodular function maximization in (Balkanski et al.,17). Although the results of this paper is somehow negative (a submodular function is hard to be minimized from samples) and thus its practical usefulness is a bit questionable, this paper could be a clue for further discussions of this type of problem, which would be important in this community.

Reviewer 2



Informally stated, the paper states the following. There exists a family F of submodular functions valued in [0, 1], so that if after polynomially many samples any algorithm estimates some set S as the minimizer, there exists some f in F that is consistent with the samples and on which S is suboptimal optimum by at least 1/2. The argument is based on the probabilistic method, i.e., by showing that such functions exist with non-vanishing probability. Moreover, the class that is considered is shown to be PAC-learnable, effectively showing that for submodular functions PAC-learnability does not imply "minimizability" from samples. However, one has to be careful about this argument. Namely, PAC learnability guarantees low error only on samples coming from the training distribution, while to minimize the function it is necessary to approximate it well near the optimum (where we might lack samples) up to an additive factor. Thus, it is not very surprising to me that the former does not imply the latter and that one can construct function classes to fool any distribution. This is a purely theoretical result that both answers an interesting question, and also provides with a construction of intuitive class of submodular functions which are (as proven) hard to minimize from samples. I only read the (sketched) proofs in the main text, which seem to be very rigorously executed. The paper is very dense, which I do not think can be easily improved on, but it is nicely written. Based on this grounds, I would recommend this paper for acceptance. I think it would be more interesting if the authors present some natural settings where the learning setting they consider is used in practice. In other words, when does learn a submodular functions from training data of the form (S_i, F(S_i))? This setting is different from how submodular functions are learned in the vision domain (where one might argue is the primary application field of submodular minimization). There, one is given a set of instances, which we want to have a low energy, either using a min-margin approach, or via maximum likelihood. Questions / Remarks: 1. The argument l129-132 is unclear to me. Do you want to say that as n -> oo both P_{f\in F}(f(S) >= 1/2) and |F'|/|F| go to one, there has to exist an f in |F'| satisfying f(S) >= 1/2 - o(1)? 2. In prop. 2 you do not define F_A(B), which I guess it means F(A \cup B) - F(B), which is confusing as it is typically defined the other way round. You could also use F(A | B), which is I guess the more commonly used notation. 3. I think the discussion after prop 2 and before section 3 should be moved somewhere else, as it does not following from the proposition itself. 4. Maybe add a note that U(A) means drawn uniformly at random from the set A. 5. l212: It is not motivated why you need one predictor for each possible masking element. 6. When you minimize a symmetric submodular function it is to minimize it over all sets different from the empty set and the ground set. This is exactly what is done using Queyranne's algorithm in cubic time. 7. Shouldn't the 1 be 1/2 in the last equation of the statement of lemma 4? 8. I find the notation in the two inequalities in Lemma 5 (after "we get") confusing. Does it mean that it holds for any h(n) = o(1) function with the given probability? Also, maybe add an "and" after the comma there. 9. l98: a word is missing after "desirable optimization"

Reviewer 3



The authors study hardness results on minimizing submodular functions form samples. The results presented in the paper are interesting and non trivial and I think that the paper can be of interest for the NIPS audience. The authors start by presenting a hardness results in the setting where the samples are drawn from a uniform distribution. The main intuition behind this result is that there exists submodular functions that achieve their minimum only when enough elements from a specific subset are sampled and this event happen only with low probability. Then the authors show how to extend this idea to prove that their construction can be extended to handle non uniform samples. Interestingly the function that they construct is learnable but not optimizable. I find this particularly interesting and surprising. Overall, I think that the paper is interesting and I suggest to accept it. Minor comment: - line 140: remove ')'